# Association of ischemic stroke onset time with presenting severity, acute progression, and long-term outcome: A cohort study

Wi-Sun Ryu[1,2], Keun-Sik Hong[3], Sang-Wuk Jeong[1], Jung E. Park[1], Beom Joon Kim[4], Joon-Tae Kim[5], Kyung Bok Lee[6], Tai Hwan Park[7], Sang-Soon Park[7], Jong-Moo Park[8], Kyusik Kang[9], Yong-Jin Cho[3], Hong-Kyun Park[3], Byung-Chul Lee[10], Kyung-Ho Yu[10], Mi Sun Oh[10], Soo Joo Lee[11], Jae Guk Kim[11], Jae-Kwan Cha[12], Dae-Hyun Kim[12], Jun Lee[13], Moon-Ku Han[4], Man Seok Park[5], Kang-Ho Choi[5], Juneyoung Lee[14], Jeffrey L. Saver[15,16], Eng H. Lo[16,17], Hee-Joon Bae[4,16☯*], Dong-Eog Kim[1,2,16☯]*

1 Department of Neurology, Dongguk University Ilsan Hospital, Goyang, Korea, 2 National Priority Research Center for Stroke, Goyang, Korea, 3 Department of Neurology, Inje University Ilsan Paik Hospital, Goyang, Korea, 4 Department of Neurology, Seoul National University Bundang Hospital, Seongnam, Korea, 5 Department of Neurology, Chonnam National University Hospital, Gwangju, Korea, 6 Department of Neurology, Soonchunhyang University Hospital, Seoul, Korea, 7 Department of Neurology, Seoul Medical Center, Seoul, Korea, 8 Department of Neurology, Uijeongbu Eulji Medical Center, Uijeongbu, Korea, 9 Department of Neurology, Nowon Eulji Medical Center, Eulji University School of Medicine, Seoul, Korea, 10 Department of Neurology, Hallym University Sacred Heart Hospital, Anyang, Korea, 11 Department of Neurology, Eulji University Hospital, Daejeon, Korea, 12 Department of Neurology, Dong-A University Hospital, Busan, Korea, 13 Department of Neurology, Yeungnam University Hospital, Daegu, Korea, 14 Department of Biostatistics, Korea University, Seoul, Korea, 15 Comprehensive Stroke Center, Department of Neurology, University of California, Los Angeles, California, United States of America, 16 Consortium International pour la Recherche Circadienne sur l'AVC (CIRCA), 17 Neuroprotection Research Laboratory, Departments of Radiology and Neurology, Massachusetts General Hospital, Harvard Medical School, Boston, Massachusetts, United States of America

☯ These authors contributed equally to this work.
* braindoc@snu.ac.kr (H-JB); dxtxok@gmail.com (D-EK)

**Data Availability Statement:** The informed consent obtained from the study participants does not allow the data to be made freely available

## Abstract

### Background

Preclinical data suggest circadian variation in ischemic stroke progression, with more active cell death and infarct growth in rodent models with inactive phase (daytime) than active phase (nighttime) stroke onset. We aimed to examine the association of stroke onset time with presenting severity, early neurological deterioration (END), and long-term functional outcome in human ischemic stroke.

### Methods and findings

In a Korean nationwide multicenter observational cohort study from May 2011 to July 2020, we assessed circadian effects on initial stroke severity (National Institutes of Health Stroke Scale [NIHSS] score at admission), END, and favorable functional outcome (3-month modified Rankin Scale [mRS] score 0 to 2 versus 3 to 6). We included 17,461 consecutive patients with witnessed ischemic stroke within 6 hours of onset. Stroke onset time was divided into 2 groups (day-onset [06:00 to 18:00] versus night-onset [18:00 to 06:00]) and

through any third party maintained public repository. However, data used for this submission can be made available upon reasonable request (stroke@stroke.or.kr) and the approval of the 'Comprehensive Registry Collaboration for Stroke in Korea' steering committee.

**Funding:** D-E.K is supported by the the National Priority Research Center Program Grant (NRF-2021R1A6A1A03038865), the Basic Science Research Program Grant (NRF-2020R1A2C3008295), and the Multi-ministry Grant for Medical Device Development (KMDF_PR_20200901_0098) of National Research Foundation, funded by the Korean government. The funders had no role in study design, data collection and analysis, decision to publish, or preparation of the manuscript.

**Competing interests:** I have read the journal's policy and the authors of this manuscript have the following competing interests: For the following roles, Dr. JS receives contracted hourly payments: Abbott / St. Jude Medical Clinical Trial Steering Committee Medtronic Clinical Trial Steering Committee BrainsGate Clinical Trial Steering Committee Stryker Clinical Trial Steering Committee Boehringer-Ingelheim (Prevention Only) Clinical Trial Steering Committee Cerenovus / Neuravi) Clinical Trial Steering Committee Phagenesis Clinical Trial Steering Committee For the following role, Dr. JS receives contracted stock options: Rapid Medical Clinical Trial Steering Committee.

**Abbreviations:** CE, cardioembolism; CRCS-K, Clinical Research Collaboration for Stroke-Korea; END, early neurological deterioration; INTERACT, Intensive Blood Pressure Reduction in an Acute Cerebral Hemorrhage Trial; LAA, large artery atherosclerosis; MRI, magnetic resonance imaging; mRS, modified Rankin Scale; NIHSS, National Institutes of Health Stroke Scale; SITS-MOST, Safe Implementation of Thrombolysis in Stroke-Monitoring Study; STEMI, ST-elevation myocardial infarction; SVO, small vessel occlusion; TIA, transient ischemic attack; TOAST, Trial of Org 10172 in Acute Stroke Treatment.

into 6 groups by 4-hour intervals. We used mixed-effects ordered or logistic regression models while accounting for clustering by hospitals. Mean age was 66.9 (SD 13.4) years, and 6,900 (39.5%) were women. END occurred in 2,219 (12.7%) patients. After adjusting for covariates including age, sex, previous stroke, prestroke mRS score, admission NIHSS score, hypertension, diabetes, hyperlipidemia, smoking, atrial fibrillation, prestroke antiplatelet use, prestroke statin use, revascularization, season of stroke onset, and time from onset to hospital arrival, night-onset stroke was more prone to END (adjusted incidence 14.4% versus 12.8%, $p = 0.006$) and had a lower likelihood of favorable outcome (adjusted odds ratio, 0.88 [95% CI, 0.79 to 0.98]; $p = 0.03$) compared with day-onset stroke. When stroke onset times were grouped by 4-hour intervals, a monotonic gradient in presenting NIHSS score was noted, rising from a nadir in 06:00 to 10:00 to a peak in 02:00 to 06:00. The 18:00 to 22:00 and 22:00 to 02:00 onset stroke patients were more likely to experience END than the 06:00 to 10:00 onset stroke patients. At 3 months, there was a monotonic gradient in the rate of favorable functional outcome, falling from a peak at 06:00 to 10:00 to a nadir at 22:00 to 02:00. Study limitations include the lack of information on sleep disorders and patient work/activity schedules.

## Conclusions

Night-onset strokes, compared with day-onset strokes, are associated with higher presenting neurologic severity, more frequent END, and worse 3-month functional outcome. These findings suggest that circadian time of onset is an important additional variable for inclusion in epidemiologic natural history studies and in treatment trials of neuroprotective and reperfusion agents for acute ischemic stroke.

## Author summary

### Why was this study done?

- The diurnal pattern in the distribution of ischemic stroke suggests an influence of circadian rhythms on stroke incidence, but it is unclear whether circadian rhythms may also affect the clinical severity of stroke, and whether time-of-day of stroke occurrence may affect acute clinical worsening after stroke onset.

### What did the researchers do and find?

- We performed a multicenter study of 17,461 consecutive patients with ischemic stroke to investigate whether time-of-day of stroke occurrence affected initial clinical severity and progressive clinical worsening within the first 72 hours after onset.

- Night-onset stroke patients were found to have greater clinical severity, and a higher likelihood of experiencing early neurologic worsening during the first 72 hours following symptom onset.

- These patients were also found to have a lower likelihood of favorable 3-month global disability outcome than day-onset stroke patients.

### What do these findings mean?

- Our large-scale clinical data suggest that there is a circadian variation in ischemic stroke progression and severity.

- These findings indicate that circadian time of stroke-onset may be an important factor to consider in future epidemiological studies and treatment trials for acute ischemic stroke.

- Further basic science and clinical investigation probing the chronobiologic mechanism of acute brain ischemia may help to identify new pharmacologic targets.

## Introduction

Despite a well-known morning increase in adverse cardiovascular events such as stroke [1,2] and myocardial infarction [3], there are only a few relatively small studies on the relationship between stroke onset time and short-term/long-term outcomes, with results being inconsistent [4–6]. In addition, these studies often did not account for subtypes of ischemic stroke, a heterogenous disease caused by different pathophysiological mechanisms, and often did not focus solely upon witnessed strokes with confirmed onset timing [7].

A recent preclinical study suggested that circadian rhythms may modulate the extent of brain ischemia and effects of treatment in acute stroke. Neuroprotective treatments reduced infarct growth in day-onset (inactive phase) rodent models of stroke (which corresponds to night-onset stroke in humans), but not in night-onset (active phase) rodent models of stroke (which corresponds to day-onset stroke in humans) [8]. Compared with active phase stroke models, inactive phase stroke models had more active cell death and infarct growth, which is the leading cause of early neurological deterioration (END) in patients with acute ischemic stroke [9,10]. To our knowledge, circadian effect on poststroke END has never been investigated yet.

In this nationwide multicenter study of 17,461 consecutive patients with witnessed ischemic stroke or transient ischemic attack (TIA), we investigated whether stroke onset time correlates with END as well as with initial neurological severity on admission and 3-month functional outcomes. Further, we explored whether these potential associations varied according to different stroke subtypes.

## Methods

### Study population

This study is reported as per the Strengthening the Reporting of Observational Studies in Epidemiology (STROBE) guideline (S1 Checklist). This study was conducted using a prospective multicenter stroke registry: Clinical Research Collaboration for Stroke-Korea (CRCS-K) [11–16]. Using a standardized protocol [17], data were collected from all patients with acute ischemic stroke or TIA who were admitted to 11 academic hospitals within 7 days of symptom

onset between May 2011 and July 2020. Inclusion criteria for this study were as follows: (1) witnessed stroke onset; (2) hospital arrival within 6 hours of onset; and (3) agreement to be monitored for the CRCS-K registry-related poststroke outcomes. Among these (*n* = 60,634), 42,903 were excluded sequentially according to the following reasons: (1) 3,375 patients refused to give research consent to being monitored for stroke outcomes; (2) 16,047 did not have the onset time information (9,922 wake-up stroke, 6,123 unwitnessed stroke, and 2 data missing); and (3) 23,481 arrived to the medical facility >6 hours of onset. The remaining 17,461 patients with witnessed ischemic stroke (*n* = 14,890) or TIA (*n* = 2,481) constituted the study cohort. A total of 1,893 (12.8%) patients were lost to follow-up: did not complete hospital visit nor reply to phone call for outcome capture. The institutional review boards of all participating centers (S1 Text) approved the study, and patients or their legally authorized representative(s) provided written informed consent.

## Clinical data collection

Using a standardized protocol [17], we collected demographic data, medication history, and details regarding vascular risk factors. Stroke subtypes were determined by consensus among experienced neurologists at each participating center, using a validated magnetic resonance imaging (MRI)-based algorithm [18] built on Trial of Org 10172 in Acute Stroke Treatment (TOAST) criteria as follows: large artery atherosclerosis (LAA), small vessel occlusion (SVO), cardioembolism (CE), undetermined, other-determined, and TIA. Admission National Institutes of Health Stroke Scale (NIHSS) score, prestroke modified Rankin Scale (mRS) score, and 3-month mRS score were collected prospectively [13,15].

## Assessment of END

Attending neurologists assessed neurological status for each patient daily. Further assessments were done whenever neurological deterioration was noticed by patients themselves, caregivers, or nurses. Physicians and nurses who assessed NIHSS score were trained and certified in a standard manner through a web-based education system (http://www.stroke-edu.or.kr/). In addition, systematic audits, including monthly monitoring and on-site visits to review medical records, were performed by the outcome adjudication committee to assure data quality. There are 5 core members in the adjudication committee: 3 neurologists (one of whom is a coauthor of this paper; TH Park), 2 research nurses, and 1 research coordinator. They do not adjudicate their own hospital patients' data. END was defined as any new neurological symptoms or signs or neurological worsening occurring within 72 hours after stroke onset, using the following criteria: (1) an increment in total NIHSS score of ≥2 points; (2) an increment in NIHSS consciousness score (1a to 1c) of ≥1; (3) an increment in NIHSS motor score (5a to 6b) ≥1; or (4) any new neurological deficit not assessed by the NIHSS [13,15,17,19]. Because of the descriptive nature of the study, we chose a low NIHSS cutoff for END to capture all possible events [20]. In addition, we selected the 72 hours criteria because ENDs beyond 72 hours after onset were less likely to be attributable to infarct growth. As previously described [13,15], the causes of END were classified as infarct growth, stroke recurrence, symptomatic hemorrhagic transformation, TIA, others, and unknown by consensus among experienced neurologists at each participating center.

## Statistical analysis

Analyses evaluated circadian effects both broadly comparing night-onset (18:00 to 06:00) versus day-onset (06:00 to 18:00) strokes and at a more granular level stratifying by 4-hour time periods of onset. Baseline characteristics in the night-onset versus day-onset stroke groups

were compared using the Student *t* test, the Wilcoxon rank-sum test, or the $\chi^2$ test according to variable types, as appropriate. Baseline characteristics among the groups stratified by the 4-hour time periods of onset were compared using one-way ANOVA or the Kruskal–Wallis test for continuous variables, and $\chi^2$ test for categorical variables, as appropriate.

Mixed-effects logistic regression models were used to investigate the association between stroke onset time and END while accounting for clustering by hospitals. The following predefined covariates that could be potentially associated with END were entered in the models: age, sex, prestroke mRS score, admission NIHSS score, previous stroke, hypertension, diabetes, hyperlipidemia, atrial fibrillation, smoking, stroke subtype, time from onset to hospital arrival, prestroke antiplatelet use, and prestroke statin use [20]. Considering seasonal variation in circadian rhythms [21], we also included stroke onset seasons as a covariate; we categorized patients into spring (March, April, and May), summer (June, July, and August), fall (September, October, and November), and winter (December, January, and February) groups.

To examine the association between stroke onset time and NIHSS score, we divided NIHSS score into 3 groups (0 to 1, 2 to 6, and ≥7), which allowed for each stratum to have a similar number of patients and performed mixed-effects ordered logistic regression analysis. In addition, we also used a mixed-effects negative binomial regression model with log link [22] to convey the sense of intergroup differences in stroke severity.

The effect of stroke onset time on dichotomized 3-month functional outcome (3-month mRS score 0 to 2 [favorable] versus 3 to 6 [unfavorable]) was also explored using a mixed-effects logistic regression model with adjustment for the same covariates in the models for stroke onset time versus END. The dichotomization was performed due to the violation of the proportional odds assumption.

As a post hoc analysis, stroke etiology-related differences in neurological severity, pathophysiological mechanisms, and circadian variation [7] were explored using statistical analyses after stratification by 3 stroke subtypes (LAA, SVO, and CE) [18]. The modifying effects of stroke subtypes were examined by entering an interaction term between stroke onset time and stroke subtypes into the model.

Sensitivity analyses were performed using the END criteria of the Safe Implementation of Thrombolysis in Stroke-Monitoring Study (SITS-MOST, increase of total NIHSS score ≥4). Moreover, to examine whether study results were confounded by changes in provision of care during off-hours, we examined the effect modification by weekdays versus weekend in the mixed-effects logistic regression model for the relationship between stroke onset time and END. The weekend included Korean national holidays as well as Saturdays and Sundays. In addition, we performed a sensitivity analysis on the relationship between stroke onset time and END after excluding a subset of patients with TIA.

The prospective analysis plan is available (S2 Text). The main analysis in the present report was consistent with the prospective analysis plan. The stroke subtype-related analyses were newly included at the data analysis stage. During the review process, we modified or added some more statistical analyses. First, we added the analysis investigating effect modification by weekdays versus weekend. Second, we categorized NIHSS score into 3 groups so as not to handle the score as a continuous variable. Third, we included stroke onset seasons as an additional covariate. Fourth, we added an additional sensitivity analysis that was performed after excluding patients with TIA. Data were analyzed using STATA software 16.0 (STATA Corp., Texas, USA). All significance tests were 2-sided, and $p < 0.05$ was considered statistically significant. In interaction analyses, which tend to have relatively low sensitivity, $p < 0.10$ was considered to indicate a potential interaction.

## Results

### Study population

A total of 60,364 patients with acute ischemic stroke or TIA were admitted within 7 days of symptom onset between May 2011 and July 2020 at the 11 participating centers. After excluding 42,903 patients based on the exclusion crteria, 17,461 patients were finally included in this study. A total of 1,893 (12.8%) patients were lost to follow-up and were excluded from the 3-month outcome analysis.

### Baseline characteristics of study population: Day-onset versus night-onset stroke

The mean age of the 17,461 patients was 66.9 (SD 13.4) years, and 6,900 (39.5%) were women. Witnessed ischemic strokes and TIAs were more common in the morning hours (S1 Fig). Compared with the day-onset stroke patients ($n$ = 12,449), the night-onset stroke patients ($n$ = 5,012) were more likely to be younger, men, and smokers, and were less likely to have hypertension (Table 1). In addition, the night-onset stroke patients were likely to visit hospital earlier, receive revascularization therapy more frequently, and have a longer door-to-needle time and door-to-puncture time than the day-onset stroke patients. Grouping of stroke onset times by 4-hour intervals showed that the onset-to-arrival time was shortest between 22:00 and 02:00; revascularization therapy was more frequently performed between 18:00 and 22:00 and between 02:00 and 06:00 than other 4-hour time periods; and door-to-needle time and door-to-puncture time were longer between 22:00 and 02:00 and between 02:00 and 06:00 than other 4-hour time periods (S1 Table).

END occurred in 2,219 (12.7%) patients. The leading cause of END was stroke progression (76.3%), followed by hemorrhagic transformation (8.9%), stroke recurrence (6.7%), unknown (4.0%), and others (2.8%). About two-thirds (65.1%) of END occurred on the first day after stroke onset, and 88.5% of END within 2 days after onset.

### Associations of night-onset strokes with higher neurological severity, more frequent END, and worse 3-month functional outcome

In the mixed-effects ordered logistic regression analysis, night-onset strokes had a higher likelihood of being more severe (adjusted common odds ratio = 1.08, 95% CI, 1.02 to 1.17). The mixed-effects negative binomial regression analysis (S2 Table) showed that night-onset strokes were associated with a higher presenting NIHSS score, compared with day-onset strokes (estimated mean NIHSS 6.1 versus 5.7; $p$ < 0.001). After adjusting for covariates, END was more prevalent in night-onset strokes than in day-onset strokes (14.4% versus 12.8%; adjusted risk difference 1.6% [95% CI 0.4 to 2.7]; $p$ = 0.006; Fig 1). This association was not modified by revascularization therapy ($p$ for interaction = 0.57). Functional outcome data at 3 months were available in 15,568 (87.2%) patients. After adjustment for covariates, night-onset strokes had a significantly lower likelihood of favorable functional outcome (76.5% versus 77.6%; adjusted odds ratio, 0.88 [95% CI 0.79 to 0.98]; $p$ = 0.03), compared with day-onset strokes.

When stroke onset times were grouped by 4-hour intervals, nighttime strokes had a higher likelihood of being more severe (Fig 2), rising from a nadir in the 06:00 to 10:00 time period to a peak in the 02:00 to 06:00 time period. Similar findings were observed in the mixed-effects negative binomial regression analysis in terms of the association between stroke onset time and admission NIHSS score (S2 Table). The 18:00 to 22:00 and 22:00 to 02:00 onset stroke patients were more likely to experience END than other time period onset stroke patients. At 3

**Table 1. Baseline characteristics.** Day-onset vs. night-onset ischemic stroke.

| | 06:00–18:00 (n = 12,449) | 18:00–06:00 (n = 5,012) | p-value |
|---|---|---|---|
| Age, years | 67.7 (13.1) | 65.1 (13.9) | <0.001 |
| Sex, women | 4,985 (40.0) | 1,915 (38.2) | 0.025 |
| Previous stroke | 2,613 (21.0) | 1,017 (20.3) | 0.30 |
| Hypertension | 8,018 (64.4) | 3,126 (62.4) | 0.01 |
| Diabetes | 3,533 (28.4) | 1,383 (27.6) | 0.30 |
| Hyperlipidemia | 3,884 (31.2) | 1,600 (31.9) | 0.35 |
| Current or recent* smoking | 4,349 (34.9) | 1,909 (38.1) | <0.001 |
| Atrial fibrillation | 3,139 (25.2) | 1,320 (26.3) | 0.12 |
| Coronary artery disease | 1,176 (9.5) | 497 (9.9) | 0.34 |
| Prestroke mRS 0 or 1 | 11,229 (90.2) | 4,546 (90.7) | 0.31 |
| Prestroke antiplatelet use | 3,623 (29.1) | 1,381 (27.6) | 0.04 |
| Prestroke statin use | 2,658 (21.4) | 1,076 (21.5) | 0.86 |
| Prestroke antihypertensive use | 6,329 (50.8) | 2,419 (48.3) | 0.002 |
| Prestroke antidiabetic use | 2,650 (21.3) | 1,032 (20.6) | 0.31 |
| Time from onset to hospital arrival, hour | 2.0 (0.9 to 3.5) | 1.4 (0.8 to 2.7) | <0.001[†] |
| Stroke subtype | | | <0.001 |
| LAA | 3,250 (26.1) | 1,217 (24.3) | |
| SVO | 1,354 (10.9) | 569 (11.4) | |
| CE | 3,097 (24.9) | 1,270 (25.3) | |
| Undetermined | 2,714 (21.8) | 1,015 (20.3) | |
| Other-determined | 355 (2.9) | 139 (2.8) | |
| TIA | 1,679 (13.5) | 802 (16.0) | |
| Revascularization therapy | 4,239 (34.1) | 1,812 (36.2) | 0.008 |
| Intravenous | 2,580 (20.7) | 1,212 (24.2) | |
| Intra-arterial | 592 (4.8) | 208 (4.2) | |
| Intravenous + intra-arterial | 1,067 (8.6) | 392 (7.8) | |
| Door-to-needle time, min | 36 (27 to 49) | 38 (28 to 52) | 0.001[†] |
| Door-to-puncture time, min | 98 (75 to 130) | 112 (89 to 146) | <0.001[†] |
| Season | | | 0.14 |
| Spring | 3,121 (25.1) | 1,242 (24.8) | |
| Summer | 3,254 (26.1) | 1,371 (27.4) | |
| Fall | 3,159 (25.4) | 1,199 (23.9) | |
| Winter | 2,915 (23.4) | 1,200 (23.9) | |
| Symptomatic hemorrhagic transformation | 161 (1.3) | 72 (1.4) | 0.46 |

Data are mean (SD), number (%), or median (interquartile range). Student *t* test was used for continuous variables, and $\chi^2$ test was used for categorical variables.

CE, cardioembolism; LAA, large artery atherosclerosis; mRS, modified Rankin Scale; NIHSS, National Institute of Health Stroke Scale; SVO, small vessel occlusion; TIA, transient ischemic attack.

*Quit smoking within 5 years of stroke onset.

[†]Rank-sum test was used.

months, a monotonic gradient in the proportions of favorable functional outcome was also noted, falling from a peak at 06:00 to 10:00 to a nadir at 22:00 to 02:00.

## Stroke subtype-specific associations of stroke onset time with neurological severity, END, and 3-month functional outcome

A total of 4,467 (25.6%) patients had LAA stroke, 1,923 (11.0%) SVO stroke, and 4,367 (25.0%) CE stroke. After the subtype stratification, the association of night-onset (versus day-onset)

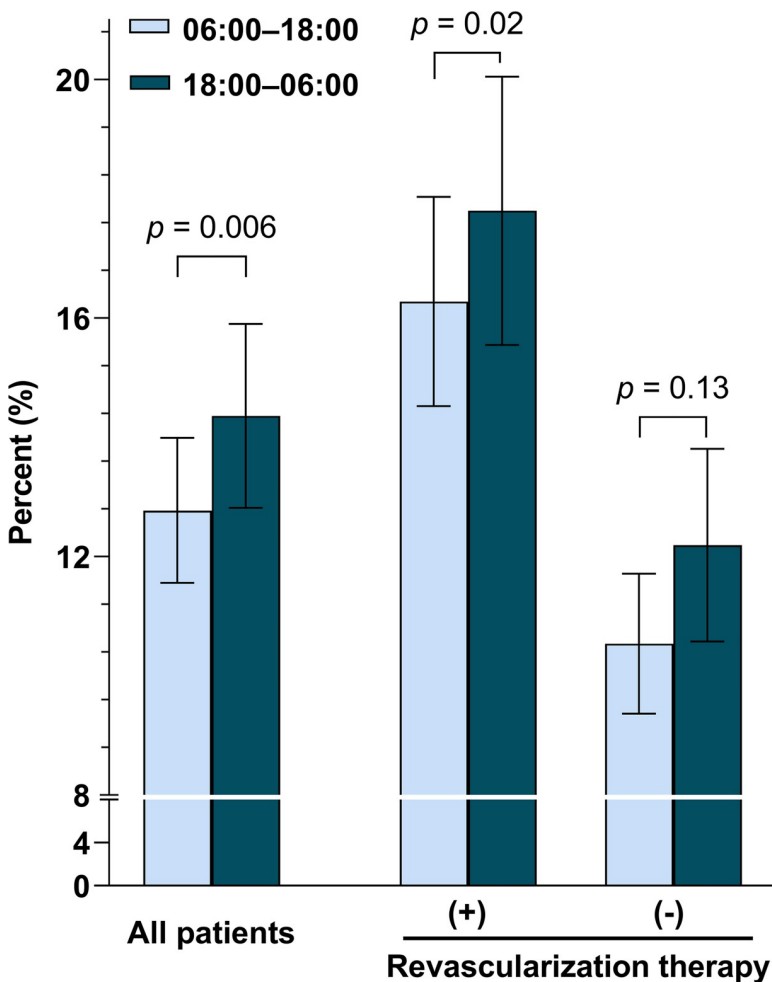

**Fig 1. Adjusted frequency of END stratified by stroke onset time (day-onset versus night-onset) and revascularization therapy.** Error bar indicates 95% confidence interval. Mixed-effects logistic regression models were used with adjustment for age, sex, previous stroke, prestroke mRS score, admission NIHSS score, hypertension, diabetes, hyperlipidemia, smoking, atrial fibrillation, prestroke antiplatelet use, prestroke statin use, revascularization, time from onset to hospital arrival, season of stroke onset, and stroke subtype. $p$ for interaction by revascularization therapy = 0.57. Note that the revascularization population has more severe stroke than the nonrevascularization population at baseline. Revascularization therapy improves their outcomes but not to the level of patients with initially milder deficits. END, early neurological deterioration; mRS, modified Rankin Scale; NIHSS, National Institutes of Health Stroke Scale.

strokes with END remained significant in LAA stroke (22.7% versus 18.7%, $p$ = 0.005), but not in SVO ($p$ = 0.13) or CE stroke ($p$ = 0.95); $p$ for interaction = 0.14 (S2 Fig). When stroke onset times were grouped by 4-hour intervals, these diurnal patterns were again most pronounced in LAA stroke, followed by SVO stroke (Table 2; see S3 Table for unadjusted comparisons). Patterns of NIHSS score variation were similar in LAA and SVO stroke, with the highest likelihood of more severe stroke in the 02:00 to 06:00 and 22:00 to 02:00 periods, respectively. CE stroke similarly had higher values at these times but also had a relatively high value in the 10:00 to 14:00, 10:00 to 14:00, and 18:00 to 22:00 periods compared with the 06:00 to 10:00 period (Table 2). In LAA stroke, END rates were higher in 18:00 to 22:00 and 22:00 to 02:00 time periods, but in SVO and CE stroke, END rates were not different across all 4-hour increments. Patterns of favorable functional outcome at 3 months were similar in LAA and CE stroke, with the lowest likelihood of favorable outcome at 22:00 to 02:00 for both subtypes. In

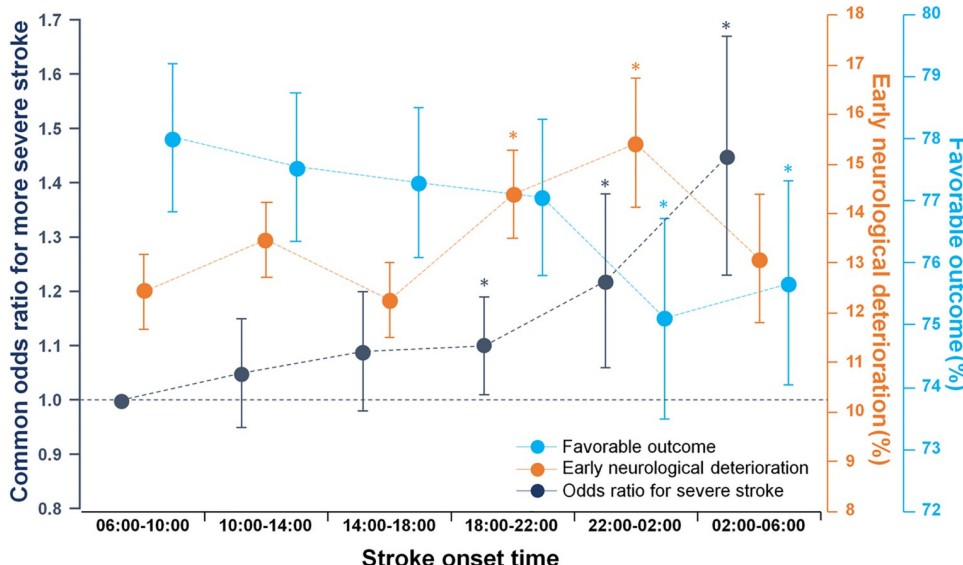

**Fig 2. Multivariable associations of stroke onset time (at 4-hour intervals) with admission NIHSS score, END, and 3-month functional outcome.** Dots and error bars indicate data estimates and their 95% confidence intervals, respectively. For the relationship between stroke onset time and presenting stroke severity, NIHSS score was stratified into 3 groups (0–1, 2–6, and ≥7) with a similar number of patients in each stratum and mixed-effects ordered logistic regression was performed with adjustment for age, sex, previous stroke, prestroke mRS score, hypertension, diabetes, hyperlipidemia, smoking, atrial fibrillation, prestroke antiplatelet use, prestroke statin use, time from onset to hospital arrival, season of stroke onset, and stroke subtype. For the relationships of stroke onset time with END and 3-month functional outcome, mixed-effects logistic regression analysis was performed with adjustment for age, sex, previous stroke, prestroke mRS score, admission NIHSS score, hypertension, diabetes, hyperlipidemia, smoking, atrial fibrillation, prestroke antiplatelet use, prestroke statin use, revascularization, time from onset to hospital arrival, season of stroke onset, and stroke subtype. *$p < 0.05$ compared with the 06:00–10:00 group. END, early neurological deterioration; mRS, modified Rankin Scale; NIHSS, National Institutes of Health Stroke Scale.

contrast, for SVO stroke, the likelihood of favorable outcome was highest in the 22:00 to 02:00 period.

## Sensitivity analyses

When END was defined using the SITS-MOST criteria [23], night-onset strokes were again more frequently associated with END (5.7%, 95% CI, 5.1% to 6.4%), compared with day-onset strokes (4.9%, 95% CI, 4.6% to 5.3%) after adjusting for covariates ($p = 0.03$; S3 Fig). There was no statistically significant weekdays versus weekend-related interaction ($p$ for interaction = 0.50; S4 Table). In addition, the sensitivity analysis for the non-TIA patients showed that night-onset stroke patients had END more frequently compared with day-onset stroke patients (S5 Table).

## Discussion

In this nationwide multicenter study on 17,461 witnessed acute ischemic stroke or TIA with the onset-to-arrival time being less than 6 hours, night-onset stroke patients had a higher admission NIHSS score, a higher likelihood of experiencing END, and a lower likelihood of favorable 3-month outcome than day-onset stroke patients. Circadian variation patterns differed among ischemic stroke subtypes, with LAA patients particularly showing more frequent END and worse 3-month functional outcome with night-onset, CE patients showing more severe presenting deficits and worse 3-month functional outcome with night-onset, and SVO

**Table 2. Multivariable associations of stroke onset time (4-hour intervals) with admission NIHSS score, neurological deterioration, and 3-month functional outcome after stratification with stroke subtypes.**

| | 06:00–10:00 | 10:00–14:00 | 14:00–18:00 | 18:00–22:00 | 22:00–02:00 | 02:00–06:00 |
|---|---|---|---|---|---|---|
| **LAA** | | | | | | |
| Number of patients | 1,047 | 1,254 | 949 | 731 | 268 | 218 |
| Admission NIHSS score* | | | | | | |
| Common odds ratios (95% CI) | Reference | 0.97 (0.83 to 1.14) | 1.13 (0.95 to 1.34) | 1.01 (0.83 to 1.22) | 1.20 (0.92 to 1.56) | 1.28 (0.96 to 1.71) |
| *p*-value | Reference | 0.73 | 0.18 | 0.92 | 0.18 | 0.09 |
| END | | | | | | |
| Adjusted incidence (95% CI)†, % | 15.7 (13.1 to 18.3) | 19.0 (16.4 to 21.6) | 15.9 (13.3 to 18.6) | 20.8 (17.4 to 24.1) | 22.1 (16.8 to 27.3) | 17.2 (12.0 to 22.3) |
| Adjusted risk difference (95% CI), % | Reference | 3.3 (0.2 to 6.4) | 0.3 (−2.9 to 3.5) | 5.1 (1.4 to 8.8) | 6.4 (0.9 to 11.9) | 1.5 (−3.9 to 6.9) |
| *p*-value | Reference | 0.038 | 0.88 | 0.006 | 0.014 | 0.58 |
| Favorable outcome‡ | | | | | | |
| Adjusted incidence (95% CI)†, % | 79.1 (75.8 to 82.3) | 78.2 (75.0 to 81.4) | 78.0 (74.6 to 81.5) | 76.7 (72.8 to 80.5) | 73.8 (68.1 to 79.5) | 75.9 (70.2 to 81.5) |
| Adjusted outcome difference (95% CI), % | Reference | −0.9 (−3.8 to 2.0) | −1.1 (−4.2 to 2.1) | −2.4 (−5.9 to 1.1) | −5.2 (−10.6 to −0.1) | −3.2 (−8.6 to 2.2) |
| *p*-value | Reference | 0.55 | 0.51 | 0.17 | 0.044 | 0.23 |
| **SVO** | | | | | | |
| Number of patients | 438 | 493 | 423 | 339 | 131 | 99 |
| Admission NIHSS score* | | | | | | |
| Common odds ratios (95% CI) | Reference | 0.87 (0.67 to 1.13) | 0.85 (0.65 to 1.12) | 0.99 (0.74 to 1.32) | 1.14 (0.76 to 1.71) | 0.94 (0.60 to 1.48) |
| *p*-value | Reference | 0.29 | 0.26 | 0.93 | 0.54 | 0.79 |
| END | | | | | | |
| Adjusted incidence (95% CI), % | 13.5 (9.2 to 17.8) | 13.8 (9.5 to 18.0) | 11.4 (7.4 to 15.4) | 16.2 (10.9 to 21.5) | 14.3 (7.3 to 21.4) | 18.8 (10.3 to 27.2) |
| Adjusted risk difference (95% CI), % | Reference | 0.3 (−4.2 to 4.7) | −2.1 (−6.5 to 2.4) | 2.7 (−2.5 to 8.0) | 0.8 (−6.3 to 8.0) | 5.3 (−3.0 to 13.5) |
| *p*-value | Reference | 0.91 | 0.36 | 0.30 | 0.82 | 0.18 |
| Favorable outcome‡ | | | | | | |
| Adjusted incidence (95% CI)†, % | 91.2 (88.0 to 94.3) | 93.1 (90.5 to 95.7) | 92.7 (89.8 to 95.6) | 93.5 (90.4 to 96.5) | 96.5 (93.2 to 99.7) | 93.0 (88.1 to 97.9) |
| Adjusted outcome difference (95% CI), % | Reference | 1.9 (−1.0 to 4.9) | 1.5 (−1.7 to 4.7) | 2.3 (−1.1 to 5.8) | 5.3 (1.2 to 9.3) | 1.8 (−3.3 to 6.9) |
| *p*-value | Reference | 0.20 | 0.35 | 0.20 | 0.048 | 0.51 |
| **CE** | | | | | | |
| Number of patients | 970 | 1,145 | 982 | 825 | 252 | 193 |
| Admission NIHSS score* | | | | | | |
| Common odds ratios (95% CI) | Reference | 1.24 (1.03 to 1.50) | 1.27 (1.05 to 1.54) | 1.25 (1.02 to 1.54) | 1.57 (1.16 to 2.12) | 1.78 (1.26 to 2.50) |
| *p*-value | Reference | 0.02 | 0.015 | 0.03 | 0.004 | 0.001 |
| END | | | | | | |
| Adjusted incidence (95% CI)†, % | 13.7 (11.3 to 16.1) | 14.6 (12.3 to 16.9) | 14.1 (11.7 to 16.5) | 14.5 (11.8 to 17.2) | 14.2 (9.7 to 18.8) | 14.8 (9.6 to 20.2) |
| Adjusted risk difference (95% CI), % | Reference | 0.9 (−2.1 to 3.8) | 0.4 (−2.7 to 3.5) | 0.8 (−2.4 to 4.1) | 0.6 (−4.4 to 5.5) | 1.2 (−4.4 to 6.7) |
| *p*-value | Reference | 0.56 | 0.79 | 0.61 | 0.82 | 0.67 |
| Favorable outcome‡ | | | | | | |
| Adjusted incidence (95% CI)†, % | 62.8 (58.8 to 66.9) | 62.0 (58.1 to 66.0) | 62.2 (58.2 to 66.2) | 59.7 (55.4 to 63.9) | 55.7 (49.7 to 61.7) | 59.8 (53.6 to 66.1) |
| Adjusted outcome difference (95% CI), % | Reference | −0.8 (−4.1 to 2.6) | −0.6 (−4.0 to 2.8) | −3.1 (−6.8 to 0.5) | −7.1 (−12.7 to −1.5) | −3.0 (−8.8 to 2.8) |
| *p*-value | Reference | 0.64 | 0.73 | 0.09 | 0.012 | 0.31 |

CE, cardioembolism; CI, confidence interval; END, early neurological deterioration; LAA, large artery atherosclerosis; mRS, modified Rankin Scale; NIHSS, National Institutes of Health Stroke Scale; SVO, small vessel occlusion.

*Admission NIHSS score was categorized into 3 groups (0–1, 2–6, and ≥7). Mixed-effects ordered logistic regression was used. Adjusted for age, sex, previous stroke, prestroke mRS score, hypertension, diabetes, hyperlipidemia, smoking, atrial fibrillation, prestroke antiplatelet use, prestroke statin use, revascularization, season of stroke onset, and time from onset to hospital arrival.

†Adjusted for age, sex, previous stroke, prestroke mRS score, admission NIHSS score, hypertension, diabetes, hyperlipidemia, smoking, atrial fibrillation, prestroke antiplatelet use, prestroke statin use, revascularization, season of stroke onset, and time from onset to hospital arrival.

‡3-month mRS score 0–2 versus 3–6 (unfavorable).

patients not evidencing night compared with day differences. To the best of our knowledge, this is the first large-scale study demonstrating circadian effects on END as well as ischemic stroke severity and 3-month functional outcome.

In a recent preclinical study using rodent models of MCA occlusion, inactive phase (daytime in rodents) onset strokes were associated with more active cell death and infarct growth, compared with active phase onset strokes [8]. The present large-scale clinical study demonstrated that inactive phase (nighttime) onset strokes in human patients were similarly associated with more severe deficits at presentation and a higher likelihood of END, which was mostly (in approximately 75%) due to infarct growth. As a possible confounder, we considered potential differences in care quality between duty hours and off-hours. However, the circadian effect on presenting stroke severity and END was not modified when stroke onset time was further stratified by weekdays versus weekend. Accordingly, the diurnal variations in severity and progression appear to be biologically driven, rather than due to variations in systems of care. Similar care quality on weekdays and weekends is in line with recent reports [24,25]. Moreover, after applying the SITS-MOST criteria of END, the independent association between stroke onset time and END was retained, which supports the robustness of our results. In addition, the incidence of END in our population (12.7%) was comparable with previous reports [26,27].

Night-onset LAA strokes were more prone to END than day-onset LAA strokes, whereas the initial neurological severity of night-onset CE strokes was higher compared with day-onset CE strokes. A possible explanation for this pattern is that LAA strokes tend to have better collateral flow than CE strokes after acute large vessel occlusion [28]. As a result, LAA stroke may have slower initial infarct growth and thus a higher likelihood of continued progression after arrival, resulting in more in-hospital END in LAA stroke. Conversely, CE stroke may have more rapid initial lesion growth, producing more neurological worsening before hospital arrival and a higher stroke severity at admission.

The current large-scale study of ischemic stroke clinical presentation and course is consonant with recent smaller studies of ischemic stroke imaging. An analysis of CT perfusion imaging data in patients with anterior large vessel occlusion stroke found larger initial ischemic core volume with night-onset compared with day-onset strokes [29]. Core growth from onset to imaging proceeded was also faster in night-onset than day-onset strokes. The present study findings in ischemic stroke align with a post hoc analysis [30] of intracerebral hemorrhage in the Intensive Blood Pressure Reduction in an Acute Cerebral Hemorrhage Trial (INTERACT) that demonstrated night-onset was associated with a higher likelihood of worse Glasgow Coma Scale score (8 or less) at admission, compared with day-onset, despite a lack of significant association with the volume of intracerebral hemorrhage [30]. These findings suggest that the pathophysiologic mechanisms of ischemic and hemorrhagic stroke are affected in a consonant manner by circadian variation.

The current study results also align with findings from investigations of myocardial ischemia. In 165 patients who had ST-elevation myocardial infarction (STEMI) with known ischemic times between 1 and 6 hours, the greatest myocardial injury occurred at 1 AM onset of ischemia and 5 AM onset of reperfusion with primary percutaneous coronary intervention [31]. A study in 1,548 consecutive patients with STEMI found poorer myocardial perfusion and larger infarct size in the early morning. As a potential causal factor, platelet aggregation was highest between 4 AM and 8 AM [32]. Other studies showed the greatest decrease in left ventricle function in myocardial infarction begin at approximately 1 AM [31,33]. These findings in ischemic heart disease are largely in line with the 4-hour interval data in the present study on ischemic stroke. Further research is required to compare the mechanisms underlying circadian variation of ischemic heart injury with those of ischemic brain injury.

This study has several strengths, including the large sample size, consecutive enrollment of patients with clear onset of ischemic stroke, prospective data collection, and regular data audits in monthly investigator meetings and annual adjudication committee meetings. END was prospectively determined during hospitalization by experienced stroke nurses and validated by the neurologists participating in the study, as part of a weekly institutional quality-of-care monitoring program for stroke patients [20]. Brain imaging was used to confirm and categorize END. There are also limitations that must be considered when interpreting our data. First, using quantitative penumbral imaging data would more directly demonstrate the circadian effects on infarct growth. Second, we did not obtain information on sleep disorders such as obstructive sleep apnea, which can affect both circadian rhythms and stroke [34]. Third, we did not obtain information on patient work/activity schedules that would allow probing of shift work effects. Fourth, further investigation with a larger sample size is required for a higher level of data granularity, which could demonstrate the association between stroke onset time and outcomes at relatively narrow time intervals. Fifth, we have followed the medical literature approach in use of the term "circadian" for a phenomenon that would be termed "diurnal" in the preclinical literature. Sixth, wake-up stroke was shown to be associated with a greater initial severity and a worse functional outcome at 3 months, compared with wakefulness stroke [5]. In another study, daytime-unwitnessed stroke patients were more likely to receive reperfusion therapy due to earlier hospital arrival after symptom recognition, compared with wake-up stroke patients [35]. Thus, the exclusion of unwitnessed stroke in the present study might have biased toward the null, potentially leading to an underestimation of the association between stroke onset time and outcomes.

In conclusion, compared with day-onset strokes, night-onset strokes are associated with more frequent END as well as higher initial neurological severity and worse 3-month functional outcome. We suggest that the circadian factor should be considered in designing future neuroprotection trials.

## Supporting information

**S1 Checklist. STROBE checklist.**
(DOCX)

**S1 Table. Baseline characteristics by stroke onset time (4-hour intervals).**
(DOCX)

**S2 Table. Mixed-effects negative binomial logistic regression analysis between admission NIHSS score and stroke onset time.** NIHSS, National Institutes of Health Stroke Scale.
(DOCX)

**S3 Table. Unadjusted associations of stroke onset time (4-hour intervals) with admission NIHSS score, neurological deterioration, and 3-month functional outcome after stratification with stroke subtypes.** NIHSS, National Institutes of Health Stroke Scale.
(DOCX)

**S4 Table. Multivariable associations between stroke onset time and END: Weekdays versus weekend. END, early neurological deterioration.**
(DOCX)

**S5 Table. Multivariable associations between stroke onset time and END after exclusion of TIA patients.** END, early neurological deterioration; TIA, transient ischemic attack.
(DOCX)

**S1 Fig. Distribution of the study population by stroke onset time.**
(DOCX)

**S2 Fig. Adjusted incidence of END stratified by stroke onset time and stroke subtype.**
END, early neurological deterioration.
(DOCX)

**S3 Fig. Multivariable associations between stroke onset time and END defined by the SITS-MOST criteria.** END, early neurological deterioration; NIHSS, National Institutes of Health Stroke Scale; SITS-MOST, Safe Implementation of Thrombolysis in Stroke-Monitoring Study.
(DOCX)

**S1 Text. List of institutional review boards.**
(DOCX)

**S2 Text. Prospective analysis plan (translated in English).**
(DOCX)

## Author Contributions

**Conceptualization:** Wi-Sun Ryu, Dong-Eog Kim.

**Data curation:** Wi-Sun Ryu, Keun-Sik Hong, Sang-Wuk Jeong, Jung E. Park, Beom Joon Kim, Joon-Tae Kim, Kyung Bok Lee, Tai Hwan Park, Sang-Soon Park, Jong-Moo Park, Kyusik Kang, Yong-Jin Cho, Hong-Kyun Park, Byung-Chul Lee, Kyung-Ho Yu, Mi Sun Oh, Soo Joo Lee, Jae Guk Kim, Jae-Kwan Cha, Dae-Hyun Kim, Jun Lee, Moon-Ku Han, Man Seok Park, Kang-Ho Choi, Hee-Joon Bae, Dong-Eog Kim.

**Formal analysis:** Wi-Sun Ryu, Juneyoung Lee, Jeffrey L. Saver, Eng H. Lo, Hee-Joon Bae, Dong-Eog Kim.

**Investigation:** Wi-Sun Ryu, Jeffrey L. Saver, Eng H. Lo, Hee-Joon Bae, Dong-Eog Kim.

**Methodology:** Wi-Sun Ryu, Beom Joon Kim, Juneyoung Lee.

**Project administration:** Sang-Wuk Jeong, Beom Joon Kim.

**Supervision:** Hee-Joon Bae, Dong-Eog Kim.

**Writing – original draft:** Wi-Sun Ryu, Keun-Sik Hong, Jung E. Park, Jeffrey L. Saver, Eng H. Lo, Hee-Joon Bae, Dong-Eog Kim.

**Writing – review & editing:** Wi-Sun Ryu, Keun-Sik Hong, Sang-Wuk Jeong, Joon-Tae Kim, Kyung Bok Lee, Tai Hwan Park, Sang-Soon Park, Jong-Moo Park, Kyusik Kang, Yong-Jin Cho, Hong-Kyun Park, Byung-Chul Lee, Kyung-Ho Yu, Mi Sun Oh, Soo Joo Lee, Jae Guk Kim, Jae-Kwan Cha, Dae-Hyun Kim, Jun Lee, Moon-Ku Han, Man Seok Park, Kang-Ho Choi, Juneyoung Lee, Jeffrey L. Saver, Eng H. Lo, Hee-Joon Bae, Dong-Eog Kim.

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
