## [Editor Report · Decision Letter 0]

24 Sep 2021

Dear Dr Ryu, 

Thank you for submitting your manuscript entitled "Circadian Time of Ischemic Stroke Onset Affects Presenting Severity, Acute Progression, and Long-term Outcome" for consideration by PLOS Medicine.

Your manuscript has now been evaluated by the PLOS Medicine editorial staff as well as by an academic editor with relevant expertise and I am writing to let you know that we would like to send your submission out for external peer review.

Please re-submit your manuscript within two working days, i.e. by Sep 28 2021 11:59PM.

Kind regards,

Callam Davidson

Associate Editor

PLOS Medicine

---

## [Decision Letter · Decision Letter 1]

27 Oct 2021

Dear Dr. Ryu,

Thank you very much for submitting your manuscript "Circadian Time of Ischemic Stroke Onset Affects Presenting Severity, Acute Progression, and Long-term Outcome" (PMEDICINE-D-21-04039R1) for consideration at PLOS Medicine. 

Your paper was evaluated by an associate editor and discussed among all the editors here. It was also discussed with an academic editor with relevant expertise, and sent to independent reviewers, including a statistical reviewer. The reviews are appended at the bottom of this email and any accompanying reviewer attachments can be seen via the link below:

[LINK]

In light of these reviews, I am afraid that we will not be able to accept the manuscript for publication in the journal in its current form, but we would like to consider a revised version that addresses the reviewers' and editors' comments. Obviously we cannot make any decision about publication until we have seen the revised manuscript and your response, and we plan to seek re-review by one or more of the reviewers. 

We expect to receive your revised manuscript by Nov 17 2021 11:59PM. Please email us (plosmedicine@plos.org) if you have any questions or concerns.

We look forward to receiving your revised manuscript. 

Sincerely,

Callam Davidson, 

PLOS Medicine

plosmedicine.org

To facilitate review, please include continuous line numbering in the margin of your manuscript.

Please revise your title according to PLOS Medicine's style. Your title must be nondeclarative and not a question. It should begin with main concept if possible. "Effect of" should be used only if causality can be inferred, i.e., for an RCT. Please place the study design (“a cohort study”) in the subtitle (ie, after a colon).

Please remove the ‘Funding’ information from your acknowledgements section and instead please enter this into the submission form under the ‘Financial Disclosure’ section.

The Data Availability Statement (DAS) requires revision. If the data are owned by a third party but freely available upon request, please note this and state the owner of the data set and contact information for data requests (web or email address). Note that a study author cannot be the contact person for the data.

Abstract Methods and Findings:

* Please ensure that all numbers presented in the abstract are present and identical to numbers presented in the main manuscript text.

* Please include the setting, year in which the study took place, length of follow up, and main outcome measures.

* Please include the important dependent variables that are adjusted for in the analyses.

Please temper the primacy claim in paragraph two of the introduction (‘To date, circadian effects on post-stroke END has never been investigated yet’) by adding ‘to our knowledge’.

Please provide the names of the institutional review boards that provided ethical approval.

Did your study have a prospective protocol or analysis plan? Please state this (either way) early in the Methods section.

Citations should appear before punctuation throughout the manuscript.

Please define "lost to follow-up" as used in this study. Other reasons for exclusion should be defined.

In Figures 1 and 2, please show the Y-axis beginning at zero. If this is not possible, please show a break in the axis.

Please remove the ‘competing interests’ and ‘author contributions’ from the end of the main text – in the event of publication, this information will be published as metadata based on your responses to the submission form questions.

Thank you for providing your STROBE checklist. Please replace the page numbers with paragraph numbers per section (e.g. "Methods, paragraph 1"), since the page numbers of the final published paper may be different from the page numbers in the current manuscript.

Comments from the reviewers:

Reviewer #1: The authors are to be commended for using a large prospective national multicenter stroke registry in South Korea to extend data on circadian patterns for stroke incidence to explore day vs night onset and adverse outcomes defined by early neurological deterioration (over 72 hours) on the popular NIHSS score and standard 90-day functional outcome on the mRS score. The rationale is that the ischemic penumbra may be more vulnerable in a 'resting state', as evident from basic science studies. The do confirm that neurological deterioration is greater with night onset, as well as variable differences in time to presentation and management, which were taken into account in various models with adjustment for multiple confounders which was offered by the large dataset involving 17,461 patients with either acute ischemic stroked or TIA out of the full registry database of 60,000+ cases who presented within 7 days. The data are interesting but concerns are raised over the approach to analysis and naturally, residual confounding.

1) my particular concern is the use of NIHSS to decimal places, which over-inflates the granularity of this measure. It is actually a categorical scale, so measures of 6.1 vs. 5.7 do not make any sense. All analyses need to be re-done accordingly.

2) while 'night' vs. 'day' can be a simple comparison, the analysis for 4 hourly time periods should be undertaken as a p for trend rather than categories against a control group. A significant time epoch against control could purely be due to chance from multiple testing

3) while adjustments have been made for hospital clustering and day of the week, there is also likely to be a seasonal component which also needs to be taken into account. Moreover, this also determines change in physiological variables, in particular systolic blood pressure but also blood glucose level, which may be particularly important on the ischemic penumbra and could be included in the models of baseline variables.

4) given variability in diagnosis (despite involvement of neurologists) and potential pathophysiology, I would like to see sensitivity analysis confined to acute ischemic stroke cases

5) Figure 1 should relate to 'frequency' rather than 'incidence'

Reviewer #2: See attachment

Michael Dewey

Reviewer #3: In their manuscript, Ryu et al. analyze the diurnal variation in a large national cohort of well defined and times stroke patients, and, importantly, have the ability to assess the severity of the stroke by outcome measures. 

 The authors confuse the difference between circadian (endogenous, and without any external time cue) and diurnal rhythms. Their data is in entrained conditions. Therefore, using the term circadian is wrong. This should be replaced by diurnal, time-of-day,... This is not a purely academic difference. Circadian refers to endogenous ("biological") effects, whereas there are a number of exogenous ("circumstances") effects, too. This study cannot resolve this dichotomy and, as the authors partly acknowledge by listing some limitations, can only point to correlation but not causation.

 It is therefore, in my view, not warranted to conclude in the Discussion that the "These associations appeared to be biologically driven." when pointing to the association between mice and human patients. Sleep between the species is very different, as are number of other characteristics important for stroke. 

 Moreover, the authors do not comment on how there might be a biased introduced by their "unwitnessed stroke" exclusion criteria that makes them exclude about the same number of patients than they included. Maybe some time of day has a lower likelyhood of strokes being witnessed. This should be discussed.

 The data is drawn throughout a period of about 9 years and there is nothing shown for seasonal effects nor discussed. This should be added to the caveats already mentioned (sleep, shift-work, chronotype).

In conclusion, I think this is an important study to make the medical community aware of daytime specific changes in disease and implications for treatment. There are a number of strengths here, but a few more caveats should be considered.

[LINK]

---

## [Decision Letter · Decision Letter 2]

17 Dec 2021

Dear Dr. Ryu,

Thank you very much for re-submitting your manuscript "Association of circadian time of ischemic stroke onset with presenting severity, acute progression, and long-term outcome: A cohort study" (PMEDICINE-D-21-04039R2) for review by PLOS Medicine.

I have discussed the paper with my colleagues and the academic editor and it was also seen again by three reviewers. I am pleased to say that provided the remaining editorial and production issues are dealt with we are planning to accept the paper for publication in the journal.

[LINK]

We look forward to receiving the revised manuscript by Dec 24 2021 11:59PM.   

Sincerely,

Callam Davidson, 

Associate Editor 

PLOS Medicine

plosmedicine.org

Requests from Editors:

Title: Please consider updating to ‘Association of ischemic stroke onset time and presenting severity, acute progression, and long-term outcome: A cohort study’.

Data Availability Statement: Please provide a contact email address for the steering committee in addition to the URL already provided.

Author Summary: Please revise your author summary in line with our guidelines (https://journals.plos.org/plosmedicine/s/revising-your-manuscript). More specifically, the summary should be non-technical and suitable for non-expert readers (including scientists and non-scientists). Avoid the use of complex terminology wherever possible and please make bullet points single sentences. Assertions of causality should also be avoided given the observational nature of the study (refer instead to associations).

Line 111: Remove ‘sample-sized’.

Line 150: Rather than providing the name of one representative institutional review board, please provide all IRB names in the supporting information and cite here (e.g. S1 Supporting Information). See here for more details regarding supporting information files (https://journals.plos.org/plosmedicine/s/supporting-information).

Line 230: Please include the original (unchanged) prospective analysis plan and relocate the details of additional or revised analyses performed during the revision process to your manuscript methods instead. 

Table 1: In the legend, please indicate the statistical test used to generate p-values. Please also present descriptive statistics consistently (e.g., seasonal data has ‘%’ in parentheses while other percentages do not). 

Figure 1: Please indicate the statistical test used in the legend. 

Table 2: Please provide the unadjusted comparisons as well as the adjusted comparisons.

Lines 415-421: Please consider whether this content would be better placed in the limitations discussion in the paragraph below. 

Comments from Reviewers:

Reviewer #1: The authors have adequately addressed Reviewer comments

Reviewer #2: The authors have dealt with all my points.

Michael Dewey

Reviewer #3: I have read the rebuttal and new manuscript of the authors. 

Specifically

- While I am wouldn't want to stop the publication of this manuscript over this technicality: Circadian is incorrect. If there are many other articles that have been published including the same mistake, it remains a mistake and should be corrected. It becomes an important distinction especially in the clinical context once the authors would want, as they do, assign a "biological driven" association. I am still unconvinced of this and really, especially in such retrospective data collection, they cannot reasonably concluded this.

- Regarding the seasonal variation: While I think the analysis the authors did make is useful, I fear I did not make myself clear: For example, seasonal changes in photoperiod lead to changes in wake-up time (see papers by Roenneberg, for example), and, thus, as a hypothesis, there should be a shift in the timing of stroke according to this. This would be another argument for a "biologically driven" phenomenon. 

I have no further comments.

[LINK]

---

## [Editor Report · Decision Letter 3]

5 Jan 2022

Dear Dr. Ryu,

Thank you very much for re-submitting your manuscript "Association of ischemic stroke onset time with presenting severity, acute progression, and long-term outcome: A cohort study" (PMEDICINE-D-21-04039R3) for review by PLOS Medicine.

Please address the remaining editorial and production issues at the end of this email. I look forward to receiving the revised manuscript by Jan 12 2022 11:59PM. 

Please email me (cdavidson@plos.org) if you have any questions or concerns.

Sincerely,

Callam Davidson, 

Associate Editor 

PLOS Medicine

plosmedicine.org

Requests from the editor:

Please revise the Author Summary as follows:

* Line 76: Please update this bullet to read 'The diurnal pattern in the distribution of ischemic stroke suggests an influence of circadian rhythms on stroke incidence, but it is unclear whether circadian rhythms may also affect the clinical severity of stroke, and whether time-of-day of stroke occurrence may affect acute clinical worsening after stroke onset.'

* Line 83: Please update 'ask' to 'investigate'.

* Line 86: Please begin a new bullet here (i.e. beginning 'Night-onset stroke patients...').

* Line 88: Please begin a new bullet here (i.e. beginning 'These patients were also found to have...').

* Lines 90-95: Please remove this bullet point as this level of detail is not required in the Author Summary. 

* Line 97: Please update 'demonstrate' to 'suggest'.

* Line 99: Please remove this bullet point as this level of detail is not required in the Author Summary.

Please update the text at lines 222-229 to read as follows: 

‘The prospective analysis plan is available (S2 Text). The main analysis in the present report was consistent with the prospective analysis plan. The stroke subtype-related analyses were newly included at the data analysis stage. During the review process, we modified or added some statistical analyses. First, we added the analysis investigating effect modification by weekdays vs. weekends. Second, we categorized NIHSS scores into 3 groups so as not to handle the scores as a continuous variable. Third, we included stroke onset seasons as an additional covariate. Fourth, we added an additional sensitivity analysis that was performed after excluding patients with TIA.'

---

## [Editor Report · Decision Letter 4]

11 Jan 2022

Dear Dr Ryu, 

On behalf of my colleagues and the Academic Editor, Dr Joshua Willey, I am pleased to inform you that we have agreed to publish your manuscript "Association of ischemic stroke onset time with presenting severity, acute progression, and long-term outcome: A cohort study" (PMEDICINE-D-21-04039R4) in PLOS Medicine.

PRESS

Sincerely, 

Callam Davidson 

Associate Editor 

PLOS Medicine